# A Quality Assurance Tool for Problem Formulation in Object Detection and Recognition

## Abstract

The effectiveness of developing computer vision systems depends on the correctness of the initial task formulation and on assessing whether the target requirements are compatible with the capabilities of the chosen base models. This paper proposes a systematic approach to early task-formulation validation that identifies negative transfer risks, architectural constraints, and logical inconsistencies before computationally expensive design and training begin. The proposed approach is general and can be applied to problems solved using direct analytical methods, classical machine learning, and modern neural networks.

First, the paper organizes task-audit metrics into three levels: data analysis (KL divergence, MMD), model analysis (linear probing, Anchor Alignment Score), and training-dynamics analysis. The toolkit is shown to function as an effective Go/No-Go filter that evaluates not only whether fine-tuning or an algorithmic solution is feasible, but also whether the task statement itself is logically coherent and physically observable.

Next, the paper discusses a fundamental shift in recognition task formulation driven by multimodal large language models (MLLMs). It examines mechanisms for projecting visual features into a semantic space (MLP projectors, C-Abstractor) and new ways to specify tasks via natural-language instructions, which helps overcome the limitations of rigid class taxonomies. To achieve robustness in challenging conditions, a hybrid "Orchestrator–Executor" architecture is proposed: an LLM serves as the strategic node (semantic context validation), while specialized CV models (e.g., YOLO, SAM) provide tactical accuracy (geometric validation). Finally, end-to-end mission quality control metrics (Mission Success Rate) are introduced to link technical performance to business requirements.

# 1 INTRODUCTION

Modern computer vision systems (CVS) demonstrate impressive results in object detection and recognition tasks. However, the critical success factor remains not so much the choice of a specific mathematical apparatus—whether direct computational methods, machine learning algorithms, or deep neural networks—but the quality of the task formulation and the compatibility of the source data with the chosen approach. In particular, neural network algorithms approximate transformations, and the result of this approximation is primarily determined by the dataset describing examples of transformations, rather than the network architecture itself Mohamadi et al. (2024).

It is essential to verify how accurately and correctly the source data regarding the background-target environment and the physically observable features of the objects of interest are formulated in the natural language of the subject domain. The first step on this path is building an ontology of the subject domain Sokolov (2022). However, this step, although foundational for a formalized task definition, is implemented not immediately, but after a large number of iterations and the accumulation of experience in using technical vision systems. Therefore, at the current stage of CVS utilization, definitions must rely on dictionary entries and phrases that are not fully refined.

Well-developed and proven generative networks, which can generate an image corresponding to a text description, are intended to verify the quality of such formulations. Using such tools, we can identify inaccuracies and ambiguities in the declarative task formulation at an early stage of CVS development. This, in turn, will help avoid the use of redundant processing algorithms and additional computational costs. It is fundamentally important to note that at this stage, not only the possibility of tuning or fine-tuning the system is checked, but the task definition itself is checked for consistency—the absence of internal contradictions, completeness of description, and physical observability of the required features.

This observation has a crucial methodological consequence: estimates obtained on one class of models (including machine learning heuristics or basic network architectures) can be extrapolated to predict the results of other solutions belonging to related classes. This enables the creation of an effective tool for validating task formulations at the earliest stages of development Sanodiya et al. (2025).

This problem becomes particularly relevant in the context of transfer learning and domain adaptation, where an incorrect assessment of the similarity between the source and target domains can lead to negative transfer, where the fine-tuned model demonstrates worse performance compared to a model trained from scratch Faure Ragani (2025).

# 2 THEORETICAL FOUNDATIONS AND CONNECTION WITH EXISTING APPROACHES

## 2.1 DIRECT METHODS, MACHINE LEARNING, AND NEURAL NETWORKS AS SOLVING TOOLS

Depending on the specific CVS, different paradigms can be applied for the solution. Direct methods rely on mathematical rules and filters rigidly defined by experts. Classical machine learning methods require manual feature engineering followed by statistical inference. Deep neural networks, in turn, extract dataset properties as a hierarchical structure of transformations.

For all three approaches, the quality of the initial task definition is critical. In direct methods, an inconsistent task will lead to algorithmic dead ends (unsolvable conditions); in machine learning, to the extraction of noise features; and in neural networks, to memorizing the dataset without generalization ability (overfitting). Although the neural network approach is computationally more expensive compared to the heuristic algorithms of classical artificial intelligence (GOFAI), it can reveal more subtle dependencies that are usually inaccessible to a human developer Li et al. (2025).

The key advantage of deep models is that they allow the evaluation of the data structure itself. Modern approaches demonstrate that identifying similarities between datasets can significantly reduce the amount of computation during training Du et al. (2024).

## 2.2 Modern Domain Adaptation Methods

Recent works (2024–2025) in domain adaptation systematize approaches aimed at eliminating domain shift. Mohamadi et al. (2024) highlight the following main directions: adversarial learning, discrepancy-based methods, multi-domain adaptation, teacher-student architectures, ensemble approaches, and methods based on Vision Language Models Mohamadi et al. (2024).

Of particular interest is the work on the unified DDASLA framework (2025), combining angular loss, Local Maximum Mean Discrepancy (LMMD), and entropy minimization. This approach demonstrates the importance of multi-level validation in model adaptation Sanodiya et al. (2025).

In the context of practical applications, a significant contribution was made by the work on Few-shot Metric Domain Adaptation (FMDA, 2024). This confirms that proper metric validation can significantly reduce the data volume requirements for fine-tuning Kudo & Iyatomi (2024).

## 3 A Systematic Approach to Task Definition Validation

### 3.1 Three-Level Structure of Audit Metrics

The proposed validation toolkit is organized as a hierarchy of three levels, each answering a specific class of questions: (1) statistical compatibility of domains, (2) suitability of model representations for the target task, and (3) dynamic consistency of the training process in the initial epochs Wang & Deng (2018).

#### 3.1.1 Level 1: Data Analysis

The first level evaluates the statistical similarity between the source and target domains using distance metrics of feature distributions extracted by a pre-trained encoder. Kullback–Leibler (KL) divergence is used as the base measure of discrepancy, quantifying the information loss when approximating the true distribution $P(x)$ with an auxiliary distribution $Q(x)$ ChatPaper Research (2025).

Formally, for the discrete case, KL-divergence is defined as:

$$D_{KL}(P \,\|\, Q) = \sum_x P(x) \log \frac{P(x)}{Q(x)}. \tag{1}$$

In the context of transfer learning, the distributions $P$ and $Q$ are interpreted as empirical feature distributions of the source and target domains; an increase in $D_{KL}$ indicates an amplification of domain shift and, as shown in several 2024 works, is a predictor of the fine-tuned model's accuracy degradation ChatPaper Research (2025).

An additional first-level metric is Maximum Mean Discrepancy (MMD), which measures the distance between the mean values of distributions in a Reproducing Kernel Hilbert Space (RKHS). For two sets of features $\{x_i\}_{i=1}^n$ and $\{y_j\}_{j=1}^m$, MMD in kernel form is written as:

$$\mathrm{MMD}^2(\mathcal{F}, X, Y) = \mathbb{E}_{x,x'} k(x, x') + \mathbb{E}_{y,y'} k(y, y') - 2\,\mathbb{E}_{x,y} k(x, y), \tag{2}$$

where $k(\cdot, \cdot)$ is a positive-definite kernel (in practice, a combination of RBF kernels with different scales) Wang & Deng (2018).

Local Maximum Mean Discrepancy (LMMD), proposed within the Deep Subdomain Adaptation Network (DSAN), refines MMD by introducing local class-wise weights: for each class $c$, the weighted kernel distance between feature subsets belonging to this class in the source and target domains is evaluated. This helps separate the global distribution shift from violations of the local class structure, better identifying cases where formally similar domains turn out to be semantically incompatible Sanodiya et al. (2025).

Crucially, empirical studies in 2024–2025 show that the correlation between "source-target" distance metrics (KL, MMD, LMMD) and the actual performance of transfer learning is not always high, especially in complex tasks with small target samples. This motivates the introduction of the second and third audit levels to supplement purely statistical distribution analysis Yang et al. (2023).

### 3.1.2 LEVEL 2: MODEL ANALYSIS

The second level evaluates the quality of representations extracted by a pre-trained backbone model via their applicability to the target classification or detection task. The baseline tool here is linear probing: all layers of the base model are frozen, fixed embeddings are extracted from target domain images, and only a linear classifier (e.g., logistic regression or one fully connected layer) is trained on top of them Shen (2024).

Let $f_\theta : \mathcal{X} \to \mathbb{R}^d$ be a pre-trained encoder, $\phi(x) = f_\theta(x)$ be an embedding, and $W \in \mathbb{R}^{K \times d}, b \in \mathbb{R}^K$ be the parameters of the linear classifier for a task with $K$ classes. Then the prediction is:

$$\hat{y} = \text{softmax}(W\phi(x) + b). \tag{3}$$

A high accuracy $\text{Acc}_{LP}$ when training only $W$ and $b$ is interpreted as evidence of good representation transferability to the target domain; low accuracy signals that the task either requires deep adaptation of all layers or is initially incorrectly formulated (features relevant to the class are absent in the data) Shen (2024).

Additionally, the Anchor Alignment Score metric can be used to quantitatively evaluate the consistency of embedding geometry with the semantic structure of the target task: for selected "anchor" points of classes, the degree of preservation of their relative distances and directions when moving to the target domain is evaluated. A low Alignment Score indicates that the model focuses on features irrelevant to the target object, even if the first-level statistical metrics look acceptable Chen et al. (2025).

### 3.1.3 LEVEL 3: TRAINING DYNAMICS ANALYSIS

The third level deals with the analysis of training trajectories in early epochs and is intended for the early detection of negative transfer. Unlike the static metrics of the first two levels, this examines the time series of loss and accuracy for several strategies Wu et al. (2024):

- fine-tuning a pre-trained model on the target domain,

- training a model "from scratch",

- deliberate overfitting on a small subset of the target dataset.

According to the Politecnico di Milano (2025) results, the negative transfer metric should compare the quality of transfer learning not with a simple baseline model, but specifically with a deliberately overfitted model on target data, taken as an "upper conservative estimate" of achievable accuracy. If the fine-tuned model consistently falls short of this overfitted benchmark, even with regularization and early stopping, negative transfer is detected, and the task falls into the No-Go zone Faure Ragani (2025).

The work on SHLPT (Similarity Heuristic Lifelong Prompt Tuning) further shows that there is no universal algorithm guaranteeing positive transfer for all tasks; a learnable task similarity metric is proposed, by which they are divided into subsets with different adaptation strategies. This aligns well with the proposed three-level audit: even with complex transfer management heuristics, the basic verification of the task definition's consistency must be performed before launching computationally expensive training Wu et al. (2024).

### 3.2 GO/NO-GO FILTER AND PRACTICAL THRESHOLDS

The combination of the three levels of metrics forms a Go/No-Go decision system that serves as a formalized "common sense filter" for defining object detection and recognition tasks. In practice, the following approximate thresholds can be set Local AI Master (2025):

**Go (development/fine-tuning):** $D_{KL} < 0.5$, MMD and LMMD are small; linear probing accuracy $\text{Acc}_{LP} > 70\%$; training dynamics monotonically improve and show no signs of persistent negative transfer.

**Caution (requires data refinement and/or adaptation):** $0.5 \leq D_{KL} \leq 1.5$, $\text{Acc}_{LP} \in [50\%, 70\%]$, MMD/LMMD indicate a substantial but not catastrophic domain shift; learning curves

show sensitivity to hyperparameters and target data volume, but the potential for improvement remains with proper adaptation.

**No-Go (inconsistent task formulation / radical revision):** $D_{KL} > 1.5$, $\text{Acc}_{LP} < 50\%$, MMD and LMMD take large values, and training dynamics show persistent negative transfer (the transfer model is consistently worse than a model trained from scratch or overfitted on a small subset of the target domain). Falling into this zone is interpreted as an indicator of an incorrect task formulation: the physical features required for the solution are either missing from the data or described semantically contradictorily, and neither direct methods, machine learning methods, nor deep neural networks can provide stable convergence to the required solution Faure Ragani (2025).

Thus, the three-level Go/No-Go filter allows not only assessing the possibility of fine-tuning a specific model, but also formally checking the task formulation itself for statistical, semantic, and dynamic consistency before beginning the costly phase of CVS design and training Local AI Master (2025).

# 4 MULTIMODAL MODELS AND THE NEW PARADIGM OF TASK FORMULATION

## 4.1 TRANSITION FROM RIGID TAXONOMIES TO LANGUAGE INSTRUCTIONS

The application of multimodal large language models (MLLMs) radically changes the fine-tuning process: from using 1–5 examples (few-shot) to completely eliminating the need for fine-tuning (zero-shot) Paul (2025).

Modern MLLMs, such as CLIP, VisualBERT, and VL-BERT, integrate visual and textual data, allowing models to process and understand multiple modalities. A key concept is representation learning, which makes it possible to create joint embeddings that capture semantic relationships between modalities Li et al. (2025).

The recent work Object-Guide CLIP (OG-CLIP, 2026) demonstrates the integration of knowledge graph-driven data augmentation using a knowledge graph of 5,000 categories and 1M image-text pairs. This approach illustrates the transition from the limited semantics of training data to enriched representations via natural language prompts Zheng (2026).

## 4.2 MECHANISMS FOR PROJECTING VISUAL FEATURES

MLP (Multi-Layer Perceptron) projectors represent the simplest mechanism for mapping visual features into the semantic space of language models. Despite their simplicity, MLP projectors demonstrate effectiveness in a number of visual grounding tasks Li et al. (2025).

C-Abstractor (Cross-modal Abstractor) presents a more complex mechanism using cross-modal attention to align and integrate information from different modalities. Cross-modal attention mechanisms allow models to focus on relevant aspects of different modalities, providing more efficient processing of multimodal data Chen et al. (2025).

Modern approaches to MLLM pre-training are optimized through various strategies Li et al. (2025):

- Contrastive Learning: the model learns to distinguish between related and unrelated image-text pairs.
- Masked Language Modeling (MLM): masking tokens in the input text with the prediction of the masked content taking visual information into account.
- Image-Text Matching: stimulating a holistic understanding of both modalities and their relationships.

The work Visual Instruction Pretraining (ViTP, 2025) proposes a new paradigm for pre-training foundation models for specialized domains. ViTP embeds a Vision Transformer (ViT) backbone into a Vision-Language Model and pre-trains it end-to-end on a rich corpus of visual instructions from target domains. The proposed Visual Robustness Learning (VRL) forces ViT to learn robust and domain-relevant features from a sparse set of visual tokens Li et al. (2025).

### 4.3 Zero-shot and Few-shot Adaptation

Research in 2024–2025 shows significant progress in zero-shot and few-shot learning for LLMs. Zero-shot learning is especially effective when working with fine-tuned LLMs, as it does not require significant resources and works well with models pre-trained on instruction datasets Paul (2025).

Few-shot learning improves model performance by adding knowledge through a few examples. For instance, GPT-4 on the MMLU benchmark demonstrates an improvement of several percentage points when moving from zero-shot to a 5-shot setup. Adding chain-of-thought can further enhance performance on complex questions Paul (2025).

Modern techniques, such as LoRA (Low-Rank Adaptation), allow fine-tuning LLMs on 100 examples at a fraction of the cost of full model training. This opens up opportunities for iterative improvement through active learning: instead of pre-labeling a large corpus, one can iteratively label small batches of the most informative examples Paul (2025).

## 5 Hybrid "Orchestrator-Executor" Architecture

### 5.1 Concept of Hybrid LLM+X Systems

The Hybrid LLM+X Architecture represents a family of system designs where LLMs are tightly integrated with domain-specific modules ("X"), such as encoders, adapters, planners, retrieval engines, symbolic reasoners, tool APIs, or hardware interfaces. These architectures exploit the generative capabilities, reasoning, and language understanding of LLMs, delegating the processing of specific modalities, numerical inference, symbolic logic, or domain knowledge tasks to specialized modules Hassouna et al. (2024).

The future is likely to combine the strengths of various paradigms orq.ai (2025):

- Graph-Conversation Hybrids: structured graphs with conversational interfaces.
- Centralized-Distributed Orchestration: centralized planning with distributed execution.
- Model-Centric and Tool-Centric Approaches: balancing LLM capabilities with specialized tools.

### 5.2 LLM as a Strategic Orchestrator

In the proposed architecture, the LLM acts as a strategic node performing the following functions orq.ai (2025):

- Semantic context validation: interpreting natural language queries and extracting semantic intent.
- Strategic planning: decomposing complex tasks into a sequence of subtasks.
- Dynamic routing: selecting appropriate execution modules based on the task context.
- Monitoring and self-correction: managing feedback loops for error correction.

The pattern of LLM-Driven Orchestration with Tool APIs, demonstrated in the Cellular-X system, shows a retrieval-augmented, tool-centric agent orchestrating document retrieval, configuration generation, and self-correction loops between the LLM and hardware interface modules Hassouna et al. (2024).

### 5.3 Specialized CV Models as Tactical Executors

Specialized computer vision models provide tactical precision through geometric validation Huang et al. (2025).

The YOLO (You Only Look Once) family of models represents highly efficient real-time object detectors. The modern YOLO-World follows the standard YOLO architecture and uses a pre-trained CLIP text encoder to encode input text, allowing the integration of textual features with visual ones via RepVL-PAN (reparameterizable vision-language path aggregation network) Li & Zhang (2025).

The recent work SAMF-YOLO (2025) proposes a self-supervised, high-precision approach integrating three key components: SONet, BFAM, and specialized attention mechanisms. The model uses contrastive learning with positive sample pairs formed by applying two different augmentations to the same source image, enabling it to learn robust representations Huang et al. (2025).

SAM (Segment Anything Model) represents a foundation model for promptable segmentation. SAM 3 (2026), evolving from SAM 2, offers Promptable Concept Segmentation (PCS), expanding segmentation capabilities to the concept level. The integration of YOLO-SAM into an end-to-end framework (2025) demonstrates the effectiveness of combining detection and segmentation for real-time applications Kirillov et al. (2023); Li & Zhang (2025).

### 5.4 Advantages of the Hybrid Approach

The hybrid architecture provides the following advantages:

- Separation of responsibilities: the LLM handles qualitative aspects, while specialized modules handle quantitative tasks Hassouna et al. (2024).
- Modular testability: each component can be tested and validated independently.
- Robustness through redundancy: semantic and geometric validation complement each other.
- Scalability: the possibility of parallel execution via multi-agent systems orq.ai (2025).

The "big.LITTLE" architecture pattern allows balancing deep reasoning (large LLMs) and routine subtasks (smaller LLMs or symbolic modules) Hassouna et al. (2024).

## 6 End-to-End Mission Quality Metrics

### 6.1 Mission Success Rate (MSR)

The proposed Mission Success Rate (MSR) metric links technical performance with business requirements. MSR is defined as the proportion of tasks completed with the required level of quality from start to finish Local AI Master (2025).

Formally:

$$\text{MSR} = \frac{N_{\text{success}}}{N_{\text{total}}} \times 100\% \tag{4}$$

where $N_{\text{success}}$ is the number of successfully completed missions, and $N_{\text{total}}$ is the total number of attempts.

A mission is considered successful if the following criteria are met:

- Semantic correctness: the LLM correctly interpreted the user's intent.
- Geometric accuracy: CV models correctly localized and classified the objects of interest.
- Time constraints: the task was completed within the specified SLA (Service Level Agreement).
- Business metrics: the result meets target KPIs (e.g., accuracy ¿ 95%, latency ¡ 100ms).

### 6.2 Multi-Level Metrics System

A comprehensive assessment of system quality requires a hierarchy of metrics Local AI Master (2025):

**Technical model performance metrics:**

- Accuracy, Precision, Recall, F1-score for classification.
- IoU (Intersection over Union), mAP (mean Average Precision) for detection.
- Dice coefficient, boundary accuracy for segmentation.

**Operational efficiency metrics:**

- Process completion time: the time to complete a task from start to finish.
- Error rates: the frequency of errors in AI-driven tasks.
- Resource utilization: the efficiency of using teams, tools, or systems Local AI Master (2025).

**Adoption and acceptance metrics:**

- Adoption rate: the percentage of users actively using the system.
- Frequency of use: the frequency of using the tool within a given timeframe.
- Task coverage: the proportion of relevant tasks processed by the AI system Local AI Master (2025).

**Qualitative impact metrics:**

- Decision Quality: improving the capabilities of strategic decision-making.
- Innovation Acceleration: the speed of developing new products or services.
- Risk Mitigation: improving the identification and prevention of potential problems.
- Competitive Advantage: differentiation in the market through AI capabilities Local AI Master (2025).

## 6.3 INTEGRATION WITH BUSINESS REQUIREMENTS

MSR provides a direct link between technical metrics and business goals, allowing Local AI Master (2025):

- Assessing the ROI (Return on Investment) of AI projects.
- Identifying bottlenecks in the production pipeline.
- Prioritizing improvements based on their impact on business metrics.
- Ensuring transparency for stakeholders.

## 7 FORECASTING REQUIRED TRAINING DATA VOLUMES

### 7.1 METHODOLOGICAL BASIS FOR DATASET SIZE ESTIMATION

The proposed task definition validation toolkit has the potential to expand beyond simple Go/No-Go assessment. The metrics obtained at the data and model analysis levels can serve as a basis for forecasting the required sizes of training datasets Kudo & Iyatomi (2024).

The work on Few-shot Metric Domain Adaptation (2024) demonstrates that with proper metric validation, significant quality improvement is possible (F1-score improvement of 11.1–29.3 points) using only 10 images per disease class from the target domain. This indicates the existence of quantitative relationships between domain similarity metrics and the data volume required for fine-tuning Kudo & Iyatomi (2024).

### 7.2 FACTORS INFLUENCING REQUIRED DATA VOLUME

Based on an analysis of current literature, the following factors can be identified:

- Degree of domain shift (KL-divergence, MMD): a greater distance between domains requires a larger volume of target data ChatPaper Research (2025).
- Quality of pre-trained representations (linear probing accuracy): high transferability of representations reduces data requirements Shen (2024).

- Complexity of the target task: the number of classes, intra-class variance, the presence of occlusions.
- Availability of few-shot/zero-shot capabilities: MLLMs radically reduce data requirements Paul (2025).

### 7.3 PROSPECTS FOR FUTURE RESEARCH

Developing formal models to forecast required data volumes based on the proposed validation metrics is of significant interest for optimizing development resources. Potential directions include Kudo & Iyatomi (2024):

- Building regression models linking domain similarity metrics with the minimum required dataset size.
- Developing active learning strategies that use validation metrics to select the most informative examples for labeling.
- Creating online adaptation methods that dynamically adjust the fine-tuning strategy based on monitoring metrics during training.

## 8 COMPARISON WITH EXISTING APPROACHES

### 8.1 ADVANTAGES OF THE PROPOSED APPROACH

Compared to existing domain adaptation and transfer learning methods, the proposed approach has the following advantages:

1. **Comprehensiveness of validation:** Unlike works focusing on specific aspects (e.g., only distance metrics of distributions ChatPaper Research (2025) or only model performance metrics Shen (2024)), the proposed approach integrates a three-level validation system covering data, model, and training dynamics analysis.

2. **Practical applicability:** The DDASLA framework (2025) demonstrates impressive results on benchmarks but requires significant computational resources for training with angular loss, LMMD, and entropy minimization. The proposed validation toolkit allows decisions to be made before costly training begins, acting as a preventive filter Sanodiya et al. (2025).

3. **Integration with modern paradigms:** Works on MLLMs (2024–2025) focus primarily on architectural innovations, while the proposed approach systematizes the methodology for evaluating the applicability of these innovations to specific tasks through task formulation validation Zheng (2026).

4. **Hybrid architecture with explicit separation of responsibilities:** Unlike monolithic solutions, the proposed "Orchestrator-Executor" architecture explicitly separates semantic and geometric validation, which increases the system's interpretability and reliability orq.ai (2025).

### 8.2 LIMITATIONS AND AREAS FOR IMPROVEMENT

1. **Need for threshold calibration:** Go/No-Go criteria require tuning for different domains and task types, which may require preliminary experiments.

2. **Computational cost of linear probing:** Although linear probing is significantly more efficient than full fine-tuning, it can become a bottleneck for very large models and datasets.

3. **Limited applicability to radically new domains:** When working with domains that differ significantly from all known ones, the predictive ability of the metrics may decrease.

## 9 CONCLUSION

This paper proposes a systematic validation tool for defining object detection and recognition tasks. It is based on a fundamental principle: the effectiveness of CVS (implemented either by direct methods, machine learning tools, or neural networks) primarily depends on data quality and the logical

consistency of initial conditions. The proposed approach guarantees that not only the possibility of a software solution or network fine-tuning is verified, but also the fundamental consistency of the task formulation itself.

The three-level system of audit metrics (data analysis, model analysis, training dynamics analysis) forms an effective Go/No-Go filter for decision-making. Integration with modern MLLMs opens up new opportunities for zero-shot and few-shot adaptation, radically reducing the requirements for labeled data volumes.

The proposed hybrid "Orchestrator-Executor" architecture ensures robustness in complex conditions by combining semantic validation through LLMs and geometric validation through specialized CV models (YOLO, SAM). The end-to-end Mission Success Rate metric links technical performance with business requirements, ensuring the transparency and controllability of AI systems.

Future research could be directed at formalizing the dependencies between validation metrics and the required training data volumes, which will enable the creation of predictive models to optimize the resources used in developing computer vision systems.

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
