# OpenReview forum: "A Quality Assurance Tool for Problem Formulation in Object Detection and Recognition"
_mathai.club/MathAI/2026/Conference — MathAI 2026 Conference Submission_

### Official Review · Reviewer_3vHw · 2026-03-11
**Review of “A Quality Assurance Tool for Problem Formulation in Object Detection and Recognition”**

**Rating:** 6
**Confidence:** 3

**Review:**

This paper proposes a systematic validation framework for defining computer vision tasks in object detection and recognition. The central idea is that many failures in computer vision systems originate not from model architecture limitations but from incorrect or inconsistent task formulation and dataset design. The authors introduce a three-level audit framework designed to validate the feasibility of a computer vision task before expensive training and model development begin.
The proposed methodology consists of three validation levels: data analysis, model analysis, and training dynamics analysis. At the data level, statistical similarity between source and target domains is evaluated using metrics such as KL-divergence and Maximum Mean Discrepancy (MMD). At the model level, the paper applies representation evaluation methods such as linear probing and Anchor Alignment Score to assess whether pre-trained embeddings are suitable for the target task. Finally, the training dynamics level analyzes learning curves to detect negative transfer early during training. Together, these metrics form a practical Go/No-Go decision system for determining whether a computer vision task formulation is feasible.
2The paper also discusses modern developments in multimodal large language models and proposes a hybrid “Orchestrator–Executor” architecture, where an LLM performs high-level reasoning and task orchestration, while specialized computer vision models such as YOLO and SAM execute geometric detection and segmentation tasks. In addition, the authors introduce a mission-level evaluation metric, Mission Success Rate (MSR), that connects technical performance metrics with business-level objectives.
The work provides an interesting conceptual framework for early validation of computer vision tasks. The proposed methodology could potentially help avoid costly model training cycles when the underlying task definition or dataset is inconsistent. The discussion of hybrid architectures combining LLM reasoning with specialized computer vision models is also relevant to current research trends.
However, the paper remains largely conceptual and lacks empirical experiments demonstrating the effectiveness of the proposed validation toolkit in practical scenarios. Quantitative experiments or case studies applying the proposed methodology to real-world datasets would significantly strengthen the contribution. In addition, the thresholds used in the Go/No-Go decision system appear heuristic and may require further validation across different application domains.
Overall, the paper presents an interesting methodological perspective on task formulation and validation in computer vision systems. While the conceptual framework is useful, stronger experimental validation would improve the impact of the work.
Strengths:
Addresses an important but often overlooked issue in computer vision: correct task formulation.
Provides a structured three-level validation framework.
Integrates ideas from domain adaptation, representation learning, and multimodal models.
Connects technical evaluation with system-level metrics such as Mission Success Rate.
Weaknesses:
Mostly conceptual with limited experimental validation.
Thresholds in the Go/No-Go filter may require empirical calibration.
Lack of real-world case studies demonstrating practical impact.

---

### Official Review · Reviewer_JRNr · 2026-03-11
**Review of 'A Quality Assurance Tool for Problem Formulation in Object Detection and Recognition'**

**Rating:** 5
**Confidence:** 4

**Review:**

The paper proposes a framework for validating computer vision task formulations before model training. The authors introduce a three-level audit system consisting of (1) statistical analysis of source-target domain compatibility (e.g., KL divergence, MMD), (2) model analysis (linear probing, Anchor Alignment Score), and (3) analysis of early training dynamics to detect negative transfer. These components are combined into a Go/No-Go decision filter intended to determine whether a task formulation is feasible before costly model development. The paper further discusses multimodal LLM-based pipelines and proposes a conceptual “Orchestrator-Executor” architecture, where an LLM performs high-level task reasoning while specialized vision models execute detection and segmentation tasks. Additionally, the authors introduce a system-level metric, Mission Success Rate (MSR), to connect technical performance with operational outcomes.

Strengths:

1. Addresses an important and often overlooked issue in computer vision development i.e incorrect task formulation and dataset incompatibility can lead to failure regardless of model architecture.

2. Structured framework: three-level audit (data, model, and training dynamics) provides a clear conceptual structure for diagnosing problems in transfer learning and domain adaptation pipelines.

3. Systems perspective: The discussion of hybrid architectures combining LLM-based reasoning with specialized vision modules reflects current trends in multimodal AI systems.

4. Practical orientation: The proposed Go/No-Go filter could potentially help practitioners avoid expensive model training when the task formulation itself is flawed.

Weaknesses:

1. Limited novelty: Most components of the framework rely on existing methods, including KL divergence, MMD/LMMD, linear probing, and analysis of training dynamics. The paper mainly combines these techniques into a conceptual workflow rather than introducing new algorithms or theoretical insights.

2. Lack of empirical validation: The paper does not include experiments, case studies, or benchmarks demonstrating the effectiveness of the proposed validation framework. Without empirical evaluation, it is difficult to assess whether the proposed Go/No-Go decision system provides reliable guidance in practice.

3. Heuristic thresholds: The thresholds proposed for the Go/No-Go filter (e.g., values of KL divergence or linear probing accuracy) appear heuristic and are not supported by experiments or statistical analysis. Providing some empirical justification or supporting analysis for these, would strengthen the proposed framework.

4. Weak mathematical contribution: Given my understanding, the technical depth of the paper is limited and mathematical elements consist primarily of standard definitions.

5. Questionable reference quality: Few references appear to come from non-peer-reviewed sources (e.g., blogs or web based technical reports).


Therefore, the paper presents an interesting conceptual perspective on task validation in computer vision systems and highlights an important practical issue. However, the contribution is primarily methodological and lacks both theoretical novelty and empirical validation.

---

### Official Review · Reviewer_i2hu · 2026-03-12
**An article with a promising idea but without specific technical details and examples**

**Rating:** 4
**Confidence:** 5

**Review:**

The main idea of the article is genuinely interesting and points toward an important conceptual direction. Multimodal large language models (MLLMs) are among the current state-of-the-art approaches in computer vision, so the topic is both timely and relevant. However, the paper lacks sufficient technical detail. In particular, it does not provide a formal definition of the problem setting, which makes it difficult to assess the exact contribution of the work and the scope of the proposed method.

The paper would be significantly stronger if the authors included at least one concrete end-to-end example illustrating how the approach is intended to work in practice. For instance, it would be helpful to specify which encoder is used at the first stage, how the components interact, and which design choices are essential to the method.

Another major weakness of the paper is the quality of the references. A substantial portion of the bibliography appears unreliable: 7 out of 21 references seem to be invalid, and several entries are presented as journal articles or technical reports even though they are actually blog posts, aggregators, or other non-scholarly sources. This seriously weakens the academic credibility of the paper.

---

### Decision · Program_Chairs · 2026-03-20

**Decision:**

Accept (Poster)

**Comment:**

Dear Author(s),

On behalf of the Program Committee of the International Conference on Mathematics of Artificial Intelligence (MathAI 2026), we are pleased to inform you that your paper has been accepted for a poster presentation at MathAI 2026.

Your paper was evaluated through a rigorous two-stage review process involving both automated screening and expert review by members of the Program Committee. The reviewers recognized the quality and contribution of your work.

Important Note: The reviewers have recommended final revisions to your manuscript before the conference. Please ensure that all reviewer comments are carefully addressed in your camera-ready version. We trust that you will complete these revisions before the conference deadlines.

Presentation details:

    Format: Poster presentation

    Mode: You may present either in person (offline) at the conference venue in Sirius, Russia, or remotely via Zoom. Please indicate your preferred mode when confirming your participation.

    Conference dates: March 30 - April 3, 2026

    Website: https://mathai.club

Next steps:

    Please confirm your participation and presentation mode by replying to this email (mathai.club@yandex.ru) no later than March 15, 2026 18:00 Moscow time.

    If you plan to attend in person, the organizing committee will provide accommodation details separately.

    Please prepare your final camera-ready manuscript according to the formatting guidelines available at https://mathai.club and upload it to OpenReview by March 15, 2026 18:00 Moscow time. Ensure that all reviewer feedback has been incorporated into this final version.

Should you have any questions regarding the program, logistics, or your presentation, please do not hesitate to contact us.

We look forward to your contribution to MathAI 2026.

With kind regards,

MathAI 2026 Program Committee
International Conference on Mathematics of Artificial Intelligence
https://mathai.club
OpenReview: https://openreview.net/group?id=mathai.club/MathAI/2026/Conference
Telegram: https://t.me/MathAI_club
Email: mathai.club@yandex.ru